# *Trichophyton indotineae*—An Emerging Pathogen Causing Recalcitrant Dermatophytoses in India and Worldwide—A Multidimensional Perspective

**DOI:** 10.3390/jof8070757

**Published:** 2022-07-21

**Authors:** Silke Uhrlaß, Shyam B. Verma, Yvonne Gräser, Ali Rezaei-Matehkolaei, Maryam Hatami, Martin Schaller, Pietro Nenoff

**Affiliations:** 1Laboratory for Medical Microbiology, 04571 Rötha OT Mölbis, Germany; S.Uhrlass@Mykologie-Experten.de; 2Nirvan Skin Clinic, Vadodara 390020, Gujarat, India; skindiaverma@gmail.com; 3Nationales Konsiliarlabor für Dermatophyten, Institut für Mikrobiologie und Infektionsmedizin, Charité–Universitätsmedizin Berlin, Corporate Member of Freie Universität Berlin and Humboldt-Universität zu Berlin, Hindenburgdamm 27, 12203 Berlin, Germany; yvonne.graeser@charite.de; 4Infectious and Tropical Diseases Research Center, Health Research Institute, Ahvaz Jundishapur University of Medical Sciences, Ahvaz 61357-15794, Iran; a.r.matehkolaie@gmail.com (A.R.-M.); mh1962244@gmail.com (M.H.); 5Department of Dermatology, Eberhard Karls University of Tübingen, Liebermeisterstr. 25, 72076 Tübingen, Germany; martin.schaller@med.uni-tuebingen.de

**Keywords:** dermatophytoses, tinea corporis, tinea cruris, tinea faciei, terbinafine resistance, Itraconazole, *Trichophyton mentagrophytes*, anthropophilic, antifungal, ITS sequencing

## Abstract

*Trichophyton (T.) indotineae* is a newly identified dermatophyte species that has been found in a near-epidemic form on the Indian subcontinent. There is evidence of its spread from the Indian subcontinent to a number of countries worldwide. The fungus is identical to genotype VIII within the *T. mentagrophytes/T. interdigitale* species complex, which was described in 2019 by sequencing the Internal Transcribed Spacer (ITS) region of ribosomal DNA of the dermatophyte. More than 10 ITS genotypes of *T. interdigitale* and *T. mentagrophytes* can now be identified. *T. indotineae* causes inflammatory and itchy, often widespread, dermatophytosis affecting the groins, gluteal region, trunk, and face. Patients of all ages and genders are affected. The new species has largely displaced other previously prevalent dermatophytes on the Indian subcontinent. *T. indotineae* has become a problematic dermatophyte due to its predominantly in vitro genetic resistance to terbinafine owing to point mutations of the squalene epoxidase gene. It also displays in vivo resistance to terbinafine. The most efficacious drug currently available for this terbinafine-resistant dermatophytoses, based on sound evidence, is itraconazole.

## 1. Introduction

There is an ongoing epidemic of dermatophytosis in India and other neighboring countries in the subcontinent. The incriminated fungus is predominantly transmitted from person to person and often leads to refractory dermatophytosis. The causative dermatophyte has replaced the anthropophilic *Trichophyton (T.) rubrum*, the erstwhile predominant dermatophyte not only responsible for tinea pedis and onychomycosis, but also for dermatophytosis involving the whole body, worldwide—including India over the past few decades [1,2,3]. The newly emerged fungus—*T. mentagrophytes* genotype VIII, now called *T. indotineae*—often causes inflammatory and pruritic forms of difficult-to-treat tinea cruris, tinea corporis, and tinea faciei (Figure 1) [4]. As a result of globalization, this new emerging pathogen has been isolated in many countries outside Asia. Infections caused by the predominantly terbinafine-resistant dermatophytes—most prominently *T. indotineae*—are now found worldwide. *T. indotineae* appears to be spreading towards Europe, with notable presence in countries including United Arab Emirates, Oman, Bahrain, [5] and Iran [6]. Within Europe, the majority of non-Indian patients with tinea caused by this species have been reported in Germany (Figure 2) [7]. Furthermore, dermatophytoses due to *T. indotineae* has been described in France [8,9], Belgium [10], Switzerland [11], Greece [12], Denmark [13], China, Australia, Canada [14,15], and recently, in Vietnam [16].

## 2. Pathogen Change from *Trichophyton rubrum* to *Trichophyton mentagrophytes*

Epidemiological studies in India have shown a trend towards an increased occurrence of *T. mentagrophytes* [17]. *T. mentagrophytes* surprisingly turned out to be the most common dermatophyte with prevalence of up to 75.9 to 77.5% [18], followed by *T. rubrum* but sometimes also by other *Trichophyton* species. At the same time, parallel to the emergence of morphologically new, therapy-refractory forms of tinea in India, a pathogen change from *T. rubrum* to *T. mentagrophytes* has evidently taken place. This dermatophyte prevails against the pathogens previously found in India—primarily *T. rubrum*—and largely displaces them as the cause of tinea cruris, tinea corporis, and tinea faciei [19].

In our own multicenter experience on tinea cruris, tinea corporis, and tinea faciei in India, *T. mentagrophytes* was detectable in 138 (92.62%) of all culture-positive skin samples. *T. rubrum*, however, was isolated in only 11 (7.38%) samples [20]. Similar results were obtained with a PCR-ELISA, where 162 of 201 samples (80.56%) were dermatophyte-positive. Of these, 151 (93.21%) were identified as *T. mentagrophytes* and 11 (6.79%) as *T. rubrum*. Both old and newer studies from India still report a mix of *Trichophyton* species associated with this outbreak. This includes a higher proportion of *T. rubrum* cases than in our series. It is possible that these other reports include misidentifications due to morphological identification of dermatophytes. Our studies were based on molecular identification using sequencing of the DNA of all dermatophyte strains isolated.

## 3. *Trichophyton mentagrophytes* Genotype VIII

Until 2007, *T. mentagrophytes* was essentially a morphologically defined species that included a large number of subtypes or variants [21]. Those included zoophilic variants like *T. mentagrophytes variatio* (*var*.) *granulosum* (from rodents, e.g., mice, rats, guinea pigs, hamsters, or from lagomorphs, e.g., rabbits), *var*. *asteroides* (from rodents), *var*. *erinacei* (from hedgehogs), and *var*. *quinckeanum* (from camels and mice), which contrasted with anthropophilic variants like *T. mentagrophytes* var. *interdigitale*, var. *goetzii* (synonym *T. krajdenii*), and var. *nodulare*. The first revision of the taxonomy and classification of dermatophytes in 2008 [22] included genetic characteristics and simplified the nomenclature. *T. mentagrophytes* var. *granulosum*, var. *asteroides*, var. *interdigitale*, var. *goetzii*, and var. *nodulare*, which showed no or only single polymorphisms in the used genetic marker (ITS region), were assigned to the species *T. interdigitale*. The differences related to the ecological niche have been neglected. *T. mentagrophytes* (*sensu stricto*), on the other hand, included only the former var. *quinckeanum* from 2008 onwards (formerly, also known as *Trichophyton sarkisovii* Ivanova & Polyakov). *T. erinacei* was also classified as a separate species based on the results.

The nomenclature of dermatophytes was revised yet again in 2017, on the basis of extended polyphasic investigations (morphology, multiple genetic markers, physiology, ecological niche, propagation form) and especially via the use of several genetic markers, which demonstrated a separation of zoophilic and anthropophilic strains within *T. interdigitale*. Furthermore, the taxonomy at the genus level was also clarified. This group of dermatophytes is currently divided into seven distinct genera and at least 56 old and new species [23]. The ITS region of the rDNA has established itself as the decisive criterion for identification, for which single nucleotide polymorphisms are crucial for classification at the species level.

As of 2017, four species, anthropophilic *T. interdigitale* and zoophilic *T. mentagrophytes, T. erinacei*, and *T. quinckeanum*, have evolved from the former variants listed above [24].

Both the variants and the species have always been difficult to identify solely on the basis of morphological characteristics [25]. The inclusion of ecological and physiological characteristics appears essential for their identification. Fortunately, with molecular methods, such as PCR-based methods, we now have a tool at hand that allows for clear differentiation. Sequencing of the Internal Transcribed Spacer (ITS) region of the rDNA also identified the new dermatophyte, first isolated in India, as ITS genotype VIII of *T. mentagrophytes* (Figure 3). 

## 4. Genotypes in the *Trichophyton mentagrophytes/Trichophyton interdigitale* Species Complex (TMTISC)

*T. mentagrophytes* genotype VIII is one of over ten known genotypes within the *T. mentagrophytes/T. interdigitale* complex (Table 1). Genotypes I to IV of the former species *T. interdigitale* were identified and described in 2010 by Heidemann et al. [27]. Genotypes I and II still belong to the anthropophilic species *T. interdigitale*, while genotypes III and IV were zoophilic. The zoophilic species *T. mentagrophytes* currently includes 11 genotypes. In addition to the ITS genotype VIII, which was first described in India, these are the other *T. mentagrophytes* genotypes: III, III*, IV, V, VII, IX, XXV, XXVI, XXVII, and XXVIII. In addition, a so-called intermediate genotype of *T. interdigitale* (II*) is repeatedly isolated. Genotype II* belongs phylogenetically to the cluster of anthropophilic strains, but shows zoophilic transmission behavior [20,28].

Fifteen more genotypes were described in the *T. mentagrophytes*/*T. interdigitale* complex in 2019 [28]. These include three ITS genotypes X, XI, and XII, within *T. interdigitale*. Twelve more genotypes XIII to XXIV in the cluster of *T. mentagrophyte*s have been described in Iran. More research, such as via multilocus Sequence Typing (MLST), microsatellite analysis coupled with epidemiological data, and antifungal susceptibility testing (AFST), seems important to characterize the multitude of genotypes in this species complex. 

## 5. *Trichophyton mentagrophytes* Genotype VIII or *Trichophyton indotineae*?

The *T. mentagrophytes* genotype VIII identified by Nenoff et al. [20] can be succinctly summed up as a ‘casualty of frequently changing classification and nomenclature of dermatophytes!’ This newly described genotype of *T. mentagrophytes* understandably created confusion because it was previously assigned to *T. interdigitale*. The first strains of Delhi [30], in which grouping was based on molecular identification using sequencing of the ITS region of fungal rDNA, were referred to as *T. interdigitale*. This was because the sequencing referred to comparison sequences stored in the database of the National Center for Biotechnology Information [NCBI] in Bethesda, Maryland, USA, which consistently classified all strains within the species complex *T. mentagrophytes*/*T. interdigitale* as *T. interdigitale* until 2016 [31]. It was only in the revised version of the taxonomy in 2017 that a distinction was made between the two different species *T. mentagrophytes* (zoophilic) and *T. interdigitale* (anthropophilic) [23]. It has therefore been our constant refrain that this latest classification should be binding and be ideally used in dermatology and mycology literature [32]. 

## 6. *Trichophyton indotineae*—A New Dermatophyte Species

The current designation of *T. mentagrophytes* genotype VIII has also recently changed [33,34]. According to the new classification of dermatophytes, genotype VIII of *T. mentagrophytes* is now classified as *T. indotineae* [35] (Figure 4). The numerous genotypes within a species complex, as seen in *T. mentagrophytes*/*T. interdigitale*, are also referred to as cryptic species or molecular siblings. It has been suggested that it makes sense to describe such genotypes as independent species only if they have significant morphological differences that have clinical significance (e.g., increased pathogenicity, changed reservoir, contagiousness, and possibly resistance to antimycotics). Since that is not the case for most genotypes, it has been suggested to designate these as mere “clonal offshoots” [36]. *T. indotineae* is morphologically (Figure 5 and Figure 6) indistinguishable from *T. mentagrophytes* but exhibits an anthropophilic instead of zoophilic transmission pattern and a high level of terbinafine resistance. Its increased virulence, observed by those who frequently treat these cases, needs to be further studied and confirmed. For the record, sequences of what became known as genotype VIII of *T. mentagrophytes* or *T. indotineae* were found for the first time in the NCBI database in 2008. One of these sequences comes from Japan, which at that time was still deposited under the species name *Arthroderma benhamiae* [37]. Another strain or sequence originated in Australia and is also deposited as *Arthroderma benhamiae* (*T. mentagrophytes*) [38]. 

## 7. Sources of Infection and Routes of Transmission of *Trichophyton indotineae*

According to current knowledge, *T. indotineae* is mainly transmitted from person to person. Spread of *T. indotineae* infection as a family case was documented in Iran [6]. None of the affected members in the family had history of travel to India, and currently, infection by the species is detected in many different provinces in Iran. However, in Germany, intra-familiar transmission was demonstrated in at least one couple living in Germany, originating from Iraq [7]. Transmission of *T. indotineae* in Germany was documented in a tinea-corporis-affected baby from Bahrain and his multiple family members, a German woman and her husband from Saudi Arabia, and from a Libyan to his female partner and their child. 

Animal infections or sources of infection are documented in just six cases (Poland, Egypt, and India) [8]. A so-called obligate “anthropozation” of this species through host adaptation, i.e., away from animals and towards humans, has also been suggested [39]. *T. indotineae* has clearly adapted excellently to the milieu on the human epidermis and is also easily transmitted to other persons directly by physical contact. However, an indirect transmission is also seen via inanimate surfaces in the living environment, such as bathrooms and lavatories, and also via contaminated bed and body linen.

## 8. Terbinafine Resistance of *Trichophyton indotineae*

The terbinafine resistance of *T. indotineae* was initially observed clinically by the fact that the dermatophytosis does not respond to treatment and worsens despite adequate oral antifungal therapy. This may be due to in vitro resistance or at least reduced in vitro sensitivity of the respective dermatophytes [40]. The treatment failure of terbinafine, however, primarily affects Indian strains of *T. indotineae* [41]. It is not possible to properly estimate the approximate timing of the mutation(s) that occurred with the emergence of *T. indotineae*. In 2017, the first reports of in vitro resistance to terbinafine in India appeared.

Up to 76% of the isolates of this species and at least 57.1% of the examined *T. rubrum* isolates were resistant to terbinafine in vitro in a multicenter study in India [42]. It is amply clear that a significant number of patients with dermatophytoses caused by *T. indotineae* no longer respond satisfactorily to topical or oral treatment with this allylamine [39]. A similar situation of treatment-refractory courses of tinea cruris and tinea corporis caused by *T. indotineae* is already being seen and must also be expected in the future in countries like Germany and others in Europe and elsewhere [7]. The challenge lies in the accurate species identification of the isolate using molecular methods or the sensitivity test to terbinafine, either with a simple breakpoint method or, better, with a standardized microdilution test according to EUCAST (European Committee on Antimicrobial Susceptibility Testing) Def. 11.0, along with a mutation analysis of the squalene epoxidase (SQLE) gene by means of sequencing. Finally, the consequent appropriate topical and oral treatment is of paramount importance (Figure 7).

## 9. Abuse of Topical Glucocorticoids and the Emergence of *Trichophyton indotineae*

There seems to be an indubitable and oft-considered temporal association of proprietary creams containing topical steroids and antifungal/antibacterial agents with this relatively sudden occurrence of previously uncommon and extensive therapy-refractory chronic widespread dermatophytoses. These creams at times contain as many as four different antimicrobial agents, with three being the norm. These include antimycotic agents (e.g., like miconazole, clotrimazole, ketoconazole, terbinafine), antibiotics (gentamicin, ofloxacin, neomycin sulfate, ornidazole) and/or antiseptics (clioquinol, chlorhexidine) [43,44,45]. The most objectionable fact is the inclusion of the super-potent clobetasol propionate, a class IV topical glucocorticoid according to Niedner [44,46]. Pharmacologically termed as ‘fixed dose combinations’ (FDC) are commonly known as combination creams or cocktail creams. Alternative topical corticosteroids in the creams include beclomethasone dipropionate, betamethasone valerate (class III topical steroid), and mometasone. Beclomethasone is gaining popularity after strong objection being raised by the Indian Association of Dermatologists, Venereologists and Leprologists against the use of clobetasol propionate in such proprietary preparations due to the lack of rationale and the potential risk of serious harm to the skin and systemic absorption. Pharmaceutical companies strategically price these preparations much lower than antifungal agents, especially the newer ones, making FDCs a popular option for use [47]. These creams are available without a doctor’s prescription, are sold over the counter (OTC), and are often recommended by pharmacists. The medical consequences of long-term unsupervised use of topical glucocorticoids, especially clobetasol propionate, are not sufficiently known among patients, pharmacists, and general practitioners in India [48]. It is gratifying that more and more Indian dermatologists, and also general practitioners, are critical of the uncontrolled use of clobetasol and other glucocorticoid combinations. There is advocacy against FDCs, and regular representations are being made to Indian drug regulatory authorities and pharmaceutical industry recommending banning such irrational FDCs, or selling them only with prescription [49].

Steroid combination creams such as those found in India are available in a multitude of countries worldwide, e.g., in the African continent, and in Arab countries. In some countries, e.g., in Germany and other European countries, it is not possible by law to obtain these without prescription, but these countries are the exception.

Unsupervised long-term use of FDCs containing clobetasol and antibiotics/antimycotics over weeks and months has been receiving increasing attention [50]. Their ready cutaneous absorption leads to systemic effects, often leading to iatrogenic Cushing’s syndrome [51]. A much earlier indication of steroid misuse/abuse is steroid-modified tinea, a term we prefer to the rather loosely used ‘tinea incognita’, as the morphology of individual lesions often changes without becoming unrecognizable. Double-edged tinea, those with thick edges, pustular lesions, and multiple coalescing annular lesions of varying sizes are tell-tale signs of topical steroid abuse in our opinion. A specific but not very common morphological form of steroid-modified dermatophytosis most often due to misuse/abuse is *tinea pseudoimbricata* [52]. Unlike tinea imbricata caused by *T. concentricum*, endemic to the Philippines and Indonesia [53], tinea pseudoimbricata appears specific to topical steroid misuse, barring rare reports of its occurrence in immunosuppressed patients due to organ transplants or advanced cancer [54,55]. Morphologically, both tinea imbricata and pseudoimbricata are fairly similar, with appearance of concentric erythematous rings. Though tinea pseudoimbricata has also been described in *T. tonsurans* infections, it is seen regularly, if not commonly, in patients of dermatophytosis caused by *T. indotineae* in India and beyond [52]. Dermatophytosis caused by *T. indotineae* often begins as tinea corporis, tinea cruris, or tinea genitalis, sometimes concurrently, as inflammatory or hyperpigmented scaly and severely itchy lesions. Sometimes the lesions are difficult to label as dermatophytosis, but eventually lesions in the groin spread posteriorly to the gluteal region and to the trunk and extremities as direct spread, leading to large lesions (Figure 8) [56]. Interestingly, tinea faciei has become much more common in infections caused by *T. indotineae*. Frequently seen consequences of misuse of potent topical steroids include striae rubra, striae albae, hypopigmentation, telangiectasias, thinning of the skin, and extensive tinea [57]. Striae can also ulcerate or appear edematous [58,59].

It is speculative to what extent topical glucocorticoids and antimycotics/antibiotics have directly or indirectly promoted the development of therapy-refractory dermatophytoses and, in particular, the adaptation of *T. indotineae*. A significant change in the skin microbiome and the local immune system is conceivable. Significant risk factors for recurrent dermatophytoses, in addition to topical steroid application, are infrequent washing of clothes, occlusive tight-fitting underwear, a positive family history of tinea, and shared use of towels and bed linen [60]. Immunological characteristics of patients affected by chronic dermatophytosis include reduced interferon (IFN) γ, reduced Th1, IL-17-positive and Th17 cells, and an impaired immune reaction (delayed type of hypersensitivity, DTH) in the intradermal test [60].

These weak/reduced immunological reactions have been recorded and published previously with other forms of human dermatophytosis in contrast to models of infection. At present, there are no unique features of the immune response to *T. indotineae*. The immunological response is similar to that occurring with other dermatophytes such as *T. rubrum* [61].

## 10. Treatment of Dermatophytoses Caused by *Trichophyton indotineae*

As mentioned earlier, a significant number of chronic recurrent dermatophytoses show no response to topical and oral terbinafine [41]. *T. indotineae* is predominantly resistant to terbinafine in vitro (Figure 9). This corresponds to the detection of one or more point mutations with amino acid substitutions at position L393F or F397L of the squalene epoxidase gene [42,62]. The drug of choice for treating dermatophytosis caused by this pathogen is itraconazole [39]. The dosage is 100 mg itraconazole twice daily for 4 to 8 weeks, and in some individuals even up to 12 weeks [41]. Recently, itraconazole manufactured with SUBA (super bioavailability) technology has been found to be efficacious [63] in 50 mg twice-a-day preparation, for the same duration as for conventional itraconazole. Other drugs like fluconazole and griseofulvin also show increased minimum inhibitory concentration (MIC) against *T. indotineae* [35]. Unfortunately, there is no breakpoint value identified, and epidemiologic cut-off value (ECOFF) is being used by some workers in scenarios with unclear correlation between the MICs determined in vitro and the clinical response [13]. Strains obtained from multiple centers from across India demonstrated increased MIC corresponding with clinical non-response to fluconazole and griseofulvin in cases of tinea corporis and tinea cruris caused by *T. indotineae*. A study by Singh et al. from the Banaras Hindu University, Varanasi, India, has also shown limited effectiveness of not only oral fluconazole, griseofulvin, and terbinafine in the current epidemic of dermatophytosis in India, but also to itraconazole [64]. This randomized pragmatic trial demonstrated that at four weeks, all drugs were similarly ineffective, with cure rates being 8% or less. However, at eight weeks, the numbers of cured patients were twenty-one for fluconazole (42%), seven for griseofulvin (14%), thirty-three for itraconazole (66%), and fourteen for terbinafine (28%) demonstrating superiority of itraconazole over fluconazole, griseofulvin and terbinafine [64].

Of note is the study by Jabet et al., based on sequences available at GeneBank [8], who analyzed geographic information of a total of 537 sequences, including temporal information for 486 sequences of *T. indotineae*, in which they found evidence of *T. indotineae* in India, Australia, Iran, and Oman even during 2004–2013, well before the epidemic was described in India [65]. However, impressive research from India is available in major dermatology, mycology, and microbiology journals after 2015.

Reduced in vitro sensitivity of strains isolated in Germany to itraconazole has been reported recently [66]. The point mutation c.1342G>A in the SQLE gene of the dermatophyte was associated with a reduced in vitro susceptibility of itraconazole. Strains of *T. mentagrophytes* ITS genotype VIII that were isolated as far back as 2011 and deposited in the strain collection of the mycological laboratory of the University Dermatological Clinic in Kiel were studied. However, the extent to which itraconazole therapy fails in vivo has not yet been ascertained by the authors. This portends a particularly problematic situation because, in case of significant in vivo therapeutic failure, we would not be left with any alternative efficacious oral antimycotic therapy for the treatment of chronic dermatophytoses caused by *T. indotineae*. The combination of oral treatment with topical antifungals is always recommended because it works synergistically. Azoles, such as clotrimazole, miconazole, luliconazole, sertaconazole, bifonazole, and eberconazole, in addition to ciclopirox and amorolfine, are available in several countries, including India. The newer topical azole antifungal, luliconazole (available in Japan and India, not marketed in Germany and Europe), is reported to have superior in vitro activity against zoophilic and anthropophilic dermatophytes [67] and especially against *T. indotineae* in India [68].

## 11. Conclusions

To conclude, the rather sudden epidemiologic shift from *T. rubrum* to *T. indotineae* causing an epidemic-like situation in countries of the Indian subcontinent, especially India, and neighboring countries like UAE, Oman, Iran, amongst others, and now found spreading to Europe, is not only a bothersome disease but has become a public health issue due to the number of individuals affected and the misery it causes. Population-based studies are needed from the Indian subcontinent to understand this disease better. Molecular diagnosis of the incriminated *T. indotineae* is essential but is not readily available to the vast majority, including large teaching hospitals. The quality of antimycotic drugs, especially itraconazole, needs to be rigorously checked, and finally, strict implementation of laws favoring prescription-only drugs, curbing over-the-counter sales of antifungal drugs, and a strict ban on the manufacture and sale of FDCs containing antifungal agents and potent topical steroids, especially clobetasol propionate, are measures that are the need of the day. Regular discussions with policy makers and bureaucrats involved in ministries of health should be encouraged to act, and plans made with their cooperation should be implemented as soon as possible to curb this menace which is sweeping through several countries and which may reach hitherto unaffected regions by the virtue of migration and tourism, which are bound to increase in the post-COVID era.

## Figures and Tables

**Figure 1 jof-08-00757-f001:**
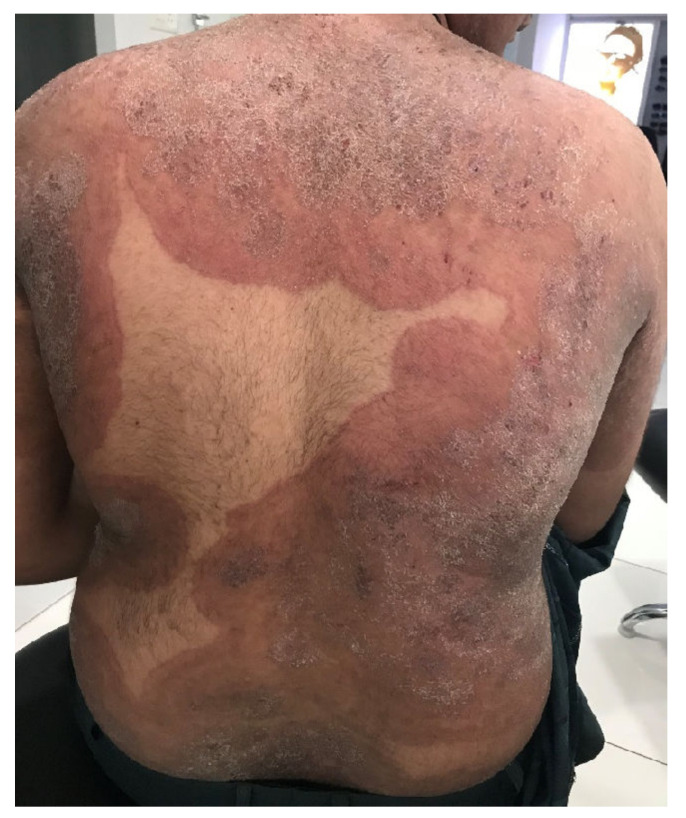
Tinea corporis generalisata in an Indian patient. The itchy erythematosquamous plaques converge over a large area and are sharply limited to the unaffected skin of the environment. Differential diagnosis includes psoriasis vulgaris, microbial eczema, or seborrheic eczema. The diagnosis can be confirmed by detection of the dermatophyte *Trichophyton mentagrophytes* genotype VIII or *Trichophyton indotineae* from skin scales. (Dr Bhavesh Devani, Drashti Skin & Eye Care Hospital-Cosmetic Laser & Hair Care Center, Rajkot, Gujarat, India).

**Figure 2 jof-08-00757-f002:**
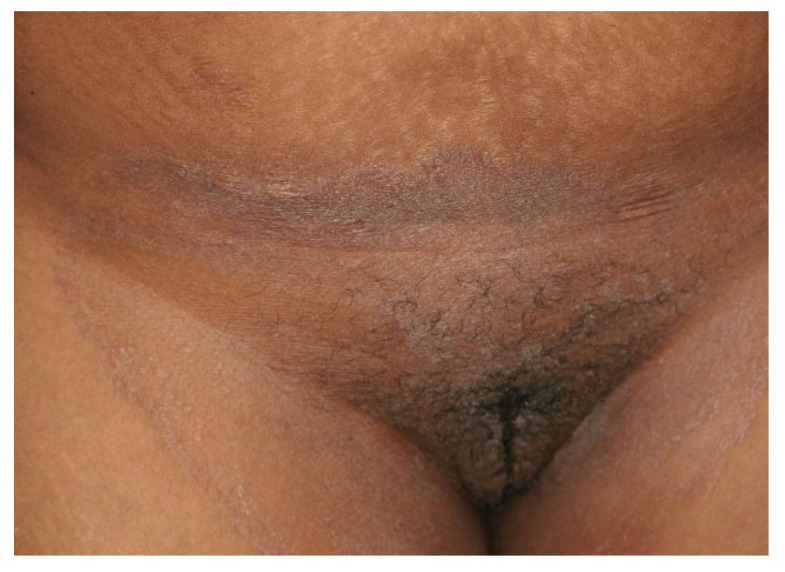
Tinea cruris and tinea genitalis in a 24-year-old patient by *Trichophyton indotineae*. (Dr. Lars Köhler, dermatologist, Mainz, Germany).

**Figure 3 jof-08-00757-f003:**
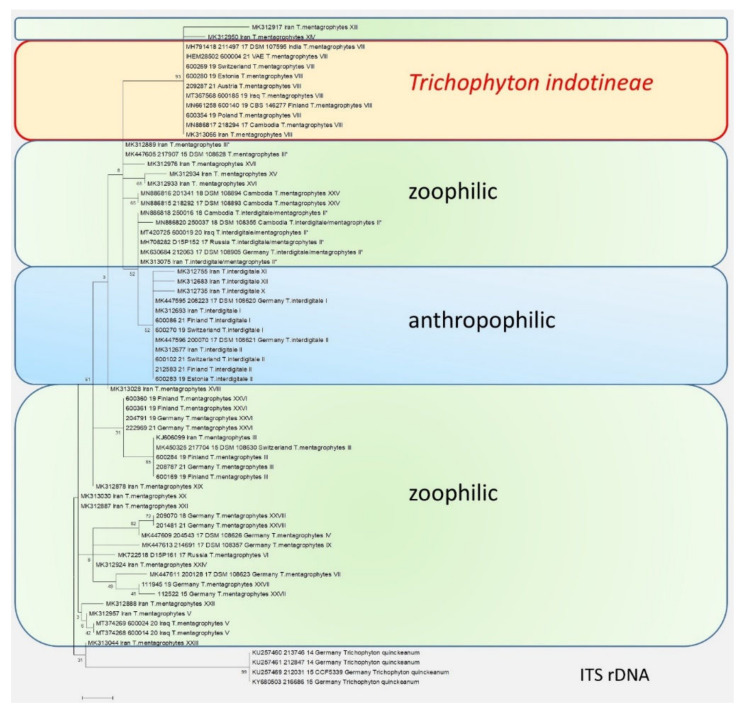
The phylogenetic analysis of the *T. mentagrophytes*/*T. interdigitale* complex based on the sequencing of the ITS regions of the rDNA. The calculations are based on the maximum likelihood method and the Tamura–Nei model from [26]. The phylogenetic family tree shows the distinction between the previously known genotypes of *T. interdigitale* and *T. mentagrophytes*, based on the sequencing of the ITS regions of rDNA genes. Genotypes I and II of the anthropophilic species *T. interdigitale* are found in the upper part of the dendrogram. Within the species *T. mentagrophytes* there are a total of 11 different genotypes—III, III*, IV, V, VII, IX, XXV, XXVI, XXVII, and XXVIII—including *T. mentagrophytes* ITS VIII (*T. indotineae*). The so-called mixed type or intermediate genotype (II*) is located between the clusters of *T. interdigitale* and *T. mentagrophytes*. The phylogenetic family tree was rooted with *Trichophyton quinckeanum*. Labeling after transmission.

**Figure 4 jof-08-00757-f004:**
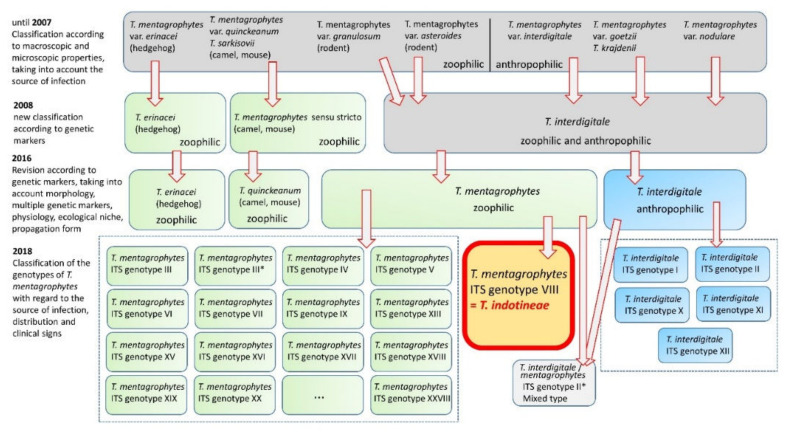
Temporal course and changes in taxonomy and nomenclature of *Trichophyton mentagrophytes* and *Trichophyton interdigitale*. The earlier zoophilic and anthropophilic variants are grouped into species over time. Within the new species, more than ten different genotypes can be distinguished. Among these, *Trichophyton mentagrophytes* ITS genotype VIII represents a clinically significant “clonal offshots” and is now considered as the independent species *Trichophyton indotineae*. In addition, there are *Trichophyton interdigitale* (anthropophilic), *Trichophyton mentagrophytes* (zoophilic), *Trichophyton erinacei* (zoophilic), *Trichophyton quinckeanum* (zoophilic), and *Trichophyton benhamiae* (zoophilic).

**Figure 5 jof-08-00757-f005:**
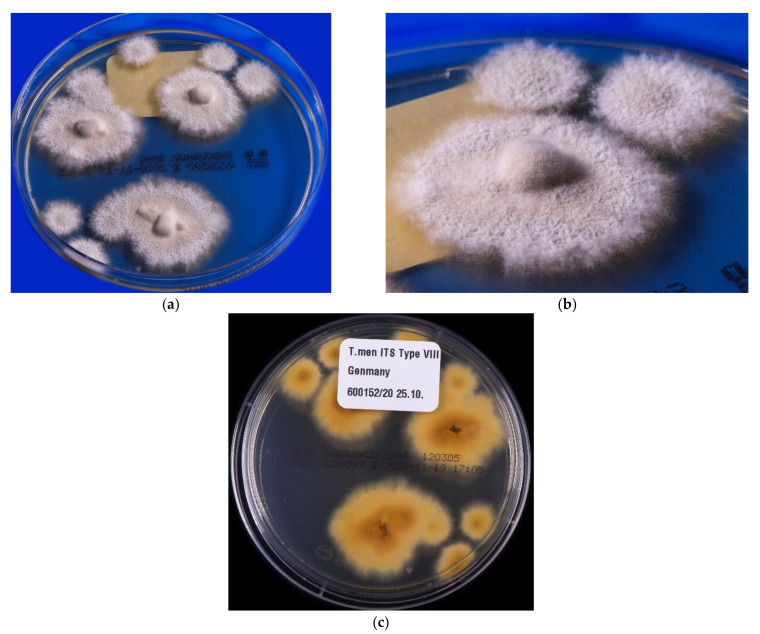
*Trichophyton indotineae*: colony growth, isolated from dandruff of a 27-year-old patient with tinea corporis. The patient comes from Bangladesh but lives and works in Germany. Growth on Sabouraud 4% glucose agar without cycloheximide additive. (**a**) Fast-growing, peripherally white, medial beige to light brown pigmented flat and granular colonies; (**b**) detailed view of colonies of the same isolate with impressive granular aspect of the thallus; (**c**) the reverse of the colonies is pigmented in light brown to yellowish.

**Figure 6 jof-08-00757-f006:**
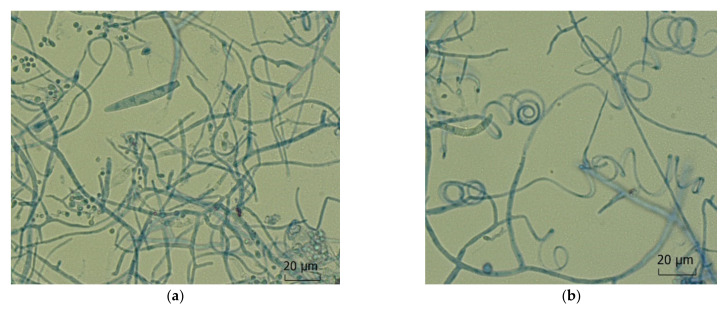
*Trichophyton indotineae*. Microscopic features of an isolate from tinea corporis and tinea genitalis of the mons pubis of a 24-year-old female from Bangladesh living in Germany. (**a**) Small and big round and oval microconidia together with spindle shaped septate macroconidia; (**b**) spiral hyphae.

**Figure 7 jof-08-00757-f007:**
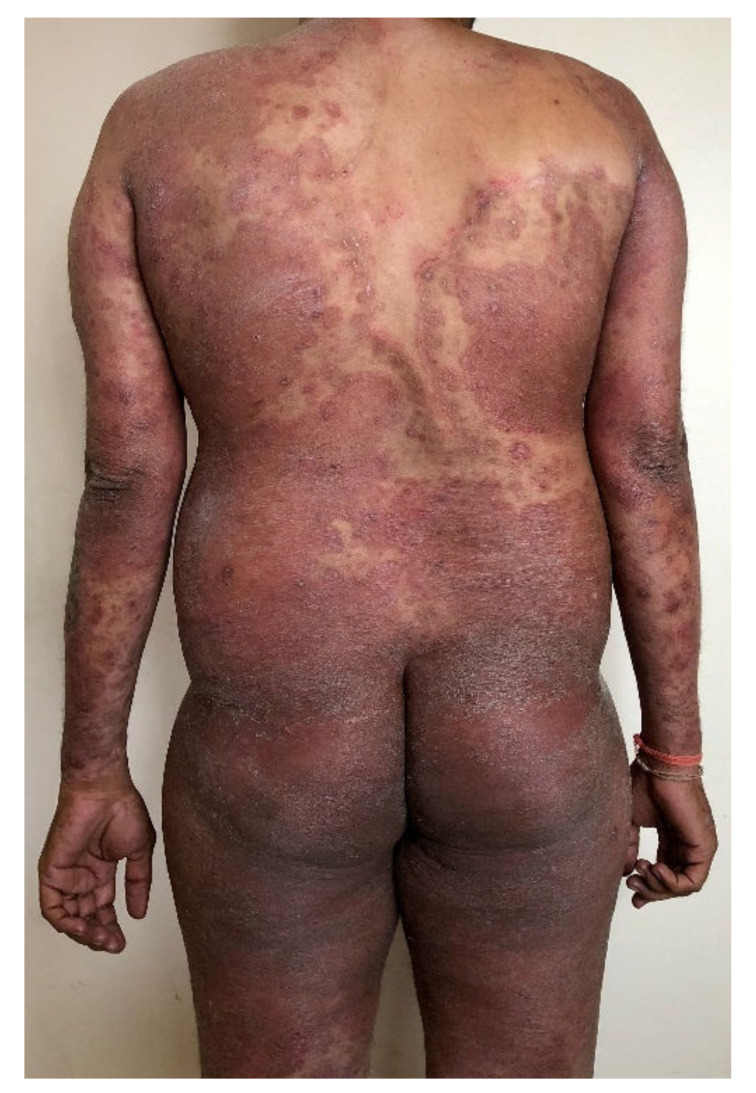
Widespread tinea generalisata (erythroderma-like) due to merging of multiple large plaques of tinea corporis in a healthy immunocompetent man in India. History of using over 100 units (tubes) of fixed-dose combination creams (FDC) containing clobetasol propionate, clotrimazole/miconazole, and gentamicin/neomycin, in addition to erratic use of topical and oral antifungal drugs, for over a year.

**Figure 8 jof-08-00757-f008:**
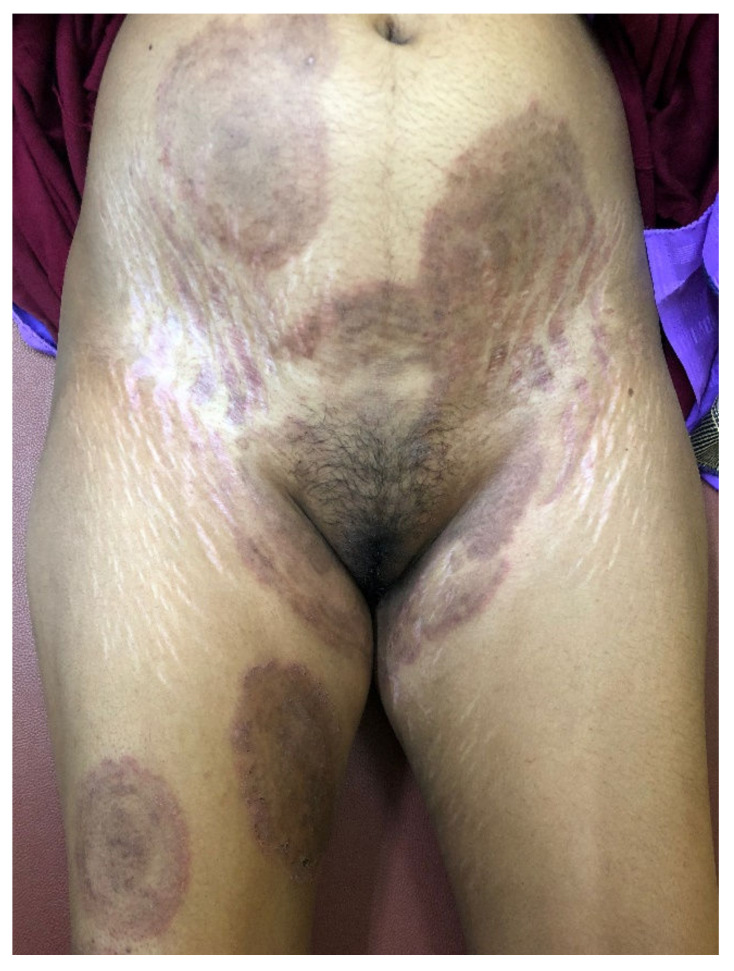
Multiple recurrent plaques of tinea corporis et cruris in a young woman applying FDCs containing the same drugs as those used for over a year by the man in Figure 7. While the plaques have partially resolved on the trunk, she is developing new lesions, including one lesion of ‘tinea pseudoimbricata’ on the right thigh with concentric circles. Some lesions of ‘tinea recidivans’ show evidence of new appearance of active inflammatory margins in healed plaques of tinea. More interestingly, and unusual for a record from a Western dermatologist, are the striking striae albae and patchy hypopigmentation, both due to the misuse of FDCs containing clobetasol propionate, itraconazole, and irrational antibiotics for several months.

**Figure 9 jof-08-00757-f009:**
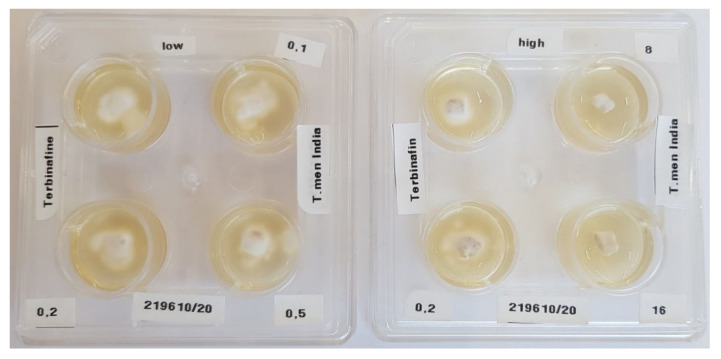
Terbinafine-resistant strain of *Trichophyton indotineae*. Isolate of a 41-year-old Indian male with tinea corporis. The susceptibility testing of this strain to terbinafine by agar dilution test yielded a minimum inhibitory concentration >0.5 μg/mL (breakpoint 0.2 μg/mL) corresponding to in vitro resistance to terbinafine. The left panel shows growing of the *T. indotineae* strain at terbinafine concentration 0.1, 0.2, and 0.5 µg/mL. The right panel shows growth of the dermatophyte at 0.2 µg/mL; however, growth was suppressed at terbinafine concentration of 8 and 16 µg/mL. The mutation analysis of the gene of squalene epoxidase revealed amino acid substitution in position F397L and thus proved terbinafine resistance.

**Table 1 jof-08-00757-t001:** Genetic origins, routes of transmission, and tinea forms of the genotypes within the *T. interdigitale*/*T. mentagrophytes complex* [28,29]. Anthropophilic genotypes (blue), mixed type (grey), zoophilic genotypes (green), and *Trichophyton indotineae* (yellow).

Genotype	MK/MF	Genetic Origin	Type of Transmission	Type of Tinea	Geographical Assignment
*Trichophyton interdigitale* I	MK312693 OM951149 OM951137 MK447595	Anthropophilic	Anthropophilic	Tinea pedis, Tinea unguium	Australia, Belgium, Finland, France, Germany, Iran, Portugal, Switzerland
*Trichophyton interdigitale* II	MK447596 MK312677 OM951151 OM951143 OM951146	Anthropophilic	Anthropophilic	Tinea pedis, Tinea unguium	Austria, Australia, Belgium, Brazil, Canada, China, Croatia, Czech Republic, Egypt, Estonia, Finland, France, Gabon, Germany, India, Iran, Japan, Malaysia, Netherlands, Portugal, Russia, South Korea, Spain, Switzerland, Tunisia, United Kingdom, USA
*Trichophyton interdigitale* X	MK312735	Anthropophilic	Anthropophilic	Tinea pedis	Iran
*Trichophyton interdigitale* XI	MK312755	Anthropophilic	Anthropophilic	Tinea manuum	Iran
*Trichophyton interdigitale* XII	MK312683	Anthropophilic	Anthropophilic	Tinea pedis	Iran
*Trichophyton interdigitale/T. mentagrophytes* II* Mixed type	MN886820 MN886818 MK313075 MK630684 MH708282 MT420725	Anthropophilic/zoophilic	Zoophilic	Tinea corporis, tinea cruris, tinea faciei, tinea unguium	Australia, Belgium, Cambodia, China, France, Germany, Greece, India, Iran, Iraq, Israel, Japan, Netherlands, New Zealand, Russia, South Korea, Thailand, United Kingdom, Vietnam
*Trichophyton mentagrophytes* III	KJ606099 MK450325 OM951159 OM951152 OM951161	Zoophilic	Zoophilic	Tinea corporis, tinea manuum	Estonia, Finland, France, Germany, Russia, Switzerland
*Trichophyton mentagrophytes* III*	MK312889 MK447605	Zoophilic	Zoophilic	Tinea faciei, tinea inguinalis, tinea corporis, tinea genitalis, tinea manuum, tinea capitis, tinea pedis	Belgium, Canada, Czech Republic, Finland, France, Germany, Greece, India, Iran, Italy, Japan, Russia, Spain, Switzerland, United Kingdom
*Trichophyton mentagrophytes* IV	MK447609	Zoophilic	Zoophilic	Tinea faciei, tinea inguinalis, tinea corporis, tinea genitalis, tinea manuum	France, Germany, Netherlands, South Africa, Switzerland, United Kingdom, USA
*Trichophyton mentagrophytes* V	MK312957 MT374268 MT374269	Zoophilic	Zoophilic	Tinea corporis, tinea capitis	Egypt, Iran, Iraq, Spain, USA
*Trichophyton mentagrophytes* VI	MK722518	Zoophilic	Zoophilic	Tinea faciei	Finland, Moldovia, Poland, Russia
*Trichophyton mentagrophytes* VII	MK447611	Zoophilic	Anthropophilic	Tinea corporis, tinea genitalis, tinea faciei, tinea cruris, tinea capitis	Australia, Austria, France, Georgia, Germany, Oman, Russia, Switzerland, Thailand, USA, Vietnam
*Trichophyton mentagrophytes* VIII = *Trichophyton indotineae*	MH791418 OM951020 OM951135 OM951144 OM951139 OM951140 MT367568 MN661258 OM951136 OM951142 OM951134 OM951138 OM951141 OM951147 MN886817 MK313066	Zoophilic	Anthropophilic	Tinea corporis, tinea cruris, tinea genitalis, tinea faciei, tinea inguinalis	Australia, Austria, Bahrain, Belgium, Cambodia, Canada, China, Denmark, Estonia, Finland, France, Germany, Greece, India, Iran, Iraq, Oman, Poland, Russia, Switzerland, UAE
*Trichophyton mentagrophytes* IX	MK447613	Zoophilic	Zoophilic	Tinea corporis	Australia, Germany
*Trichophyton mentagrophytes* XIII	MK312917	Anthropophilic	Zoophilic	Tinea corporis	Iran
*Trichophyton mentagrophytes* XIV	MK312950	Anthropophilic	Zoophilic	Tinea faciei	Iran
*Trichophyton mentagrophytes XV*	MK312934	Anthropophilic	Zoophilic	Tinea corporis, tinea faciei	Iran
*Trichophyton mentagrophytes* XVI	MK312933	Anthropophilic	Zoophilic	Tinea corporis	Iran
*Trichophyton mentagrophytes* XVII	MK312976	Anthropophilic	Zoophilic	Tinea corporis, tinea capitis, tinea cruris	Iran
*Trichophyton mentagrophytes* XVIII	MK313028	Anthropophilic	Zoophilic	Tinea corporis	Iran
*Trichophyton mentagrophytes* XIX	MK312878	Anthropophilic	Zoophilic	Tinea cruris	Iran
*Trichophyton mentagrophytes* XX	MK313030	Anthropophilic	Zoophilic	Tinea manuum	Iran
*Trichophyton mentagrophytes* XXI	MK312887	Anthropophilic	Zoophilic	Tinea corporis	Iran
*Trichophyton mentagrophytes* XXII	MK312888	Anthropophilic	Zoophilic	Tinea corporis	Iran
*Trichophyton mentagrophytes* XXIII	MK313044	Anthropophilic	Zoophilic	Tinea pedis	Iran
*Trichophyton mentagrophytes* XXIV	MK312924	Anthropophilic	Zoophilic	Tinea manuum	Belgium, Brazil, Finland, France, Germany, Iran, Japan
*Trichophyton mentagrophytes* XXV	MN886815 MN886816	Zoophilic	Zoophilic	Tinea corporis	Cambodia
*Trichophyton mentagrophytes* XXVI	OM951150 OM951153 OM951145 OM951148 OM951156	Zoophilic	Zoophilic	Tinea corporis	Finland, Germany
*Trichophyton mentagrophytes* XXVII	OM951158 OM951160	Zoophilic	Zoophilic	Tinea corporis	Germany
*Trichophyton mentagrophytes* XXVIII	OM951157 OM951162	Zoophilic	Zoophilic	Tinea corporis	Germany
*Trichophyton quinckeanum*	KU257460 KU257461 KU257469 KY680503	Zoophilic	Zoophilic	Tinea faciei, tinea corporis, tinea genitalis, tinea manuum, tinea capitis	Germany

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
