# Peer review of "Trichophyton indotineae—An Emerging Pathogen Causing Recalcitrant Dermatophytoses in India and Worldwide—A Multidimensional Perspective"

_jof, 2022, doi:10.3390/jof8070757_

Round 1

Reviewer 1 Report

Very nice article by Uhrlass et al. about T. indotinae. The tex is well written and very instructive. Worthwhile to get published
Some minor comments :
•    The end of this sentence is awkward and needs to be rephrased : "In our own multicenter experience on tinea cruris, tinea corporis and tinea faciei in India, T. mentagrophytes was detectable in 138 (92.62 %) of all culture-positive skin sam-81ples, but only in 11 (7.38 %) T. rubrum"
•    "(from rodents, e.g., rabbits)" : rabbits are not rodents, but lagomorphs
•    Some redundancies between paragraphes 3 and 4
•    Can you explain in the legend why two panels in Figure 9 ?
•    Please add the following reference to illustrate the emergence of resistance in Europe : PMID: 35330222

Author Response

Dear Reviewer,

Thank you very much for your reviews and the important hints and correction for improving the quality of our manuscript on Trichophyton intotineae.

Here our answers (in red):

Reviewer 1

Very nice article by Uhrlass et al. about T. indotinae. The tex is well written and very instructive. Worthwhile to get published

Some minor comments:
•    The end of this sentence is awkward and needs to be rephrased : "In our own multicenter experience on tinea cruris, tinea corporis and tinea faciei in India, T. mentagrophytes was detectable in 138 (92.62 %) of all culture-positive skin sam-81ples, but only in 11 (7.38 %) T. rubrum"

The sentence has been corrected and changed, see in manuscript the in red written passages.

  • "(from rodents, e.g., rabbits)": rabbits are not rodents, but lagomorphs

I have changed the description of the pets!

Those included zoophilic variants like T. mentagrophytes variatio (var.) granulosum (from rodents, e.g. mice, rats, guinea pigs, hamsters, or from lagomorphs e.g., rabbits), var. asteroides (from rodents), var. erinacei (from hedgehogs), var. quinckeanum (from camels and mice) which contrasted with anthropophilic variants like T. mentagrophytes var. interdigitale, var. goetzii (synonym T. krajdenii) and var. nodulare.

  • Some redundancies between paragraphs 3 and 4

We have checked it.

  • Can you explain in the legend why two panels in Figure 9?

The legend of the picture is now better explained, see in the manuscript (in red).

  • Please add the following reference to illustrate the emergence of resistance in Europe:

Trichophyton indotineae – an Emerging Pathogen causing recalcitrant dermatophytoses in India and worldwide - a multidimensional perspective

Authors

Silke Uhrlaß , Shyam B. Verma , Yvonne Gräser , Ali Rezaei‐Matehkolaei , Maryam Hatami , Martin Schaller , Pietro Nenoff *

This paper under submission is ours?!

Reviewer 2 Report

A very interesting article that I read in one breath due to my personal scientific interests. The manuscript is clearly written and shows a good level of language. The layout of the content is correct and all the figures significantly improve the reception of the content. It seems to me that it is possible to combine Figures 1, 2, 7 and 8 into a single figure with the notations a, b, c and d. In fact, it is a clinical picture in different forms each time. Moreover, figures 3 and 4 have to be enlarged, in their present form they are completely illegible.

Author Response

Dear Reviewer,

Thank you very much for your reviews and the important hints and correction for improving the quality of our manuscript on Trichophyton intotineae.

Here our answers (in red):

Reviewer 2

Formularende

Formularbeginn

A very interesting article that I read in one breath due to my personal scientific interests. The manuscript is clearly written and shows a good level of language. The layout of the content is correct and all the figures significantly improve the reception of the content. It seems to me that it is possible to combine Figures 1, 2, 7 and 8 into a single figure with the notations a, b, c and d. In fact, it is a clinical picture in different forms each time. Moreover, figures 3 and 4 have to be enlarged, in their present form they are completely illegible.

Yes, we agree with this ´suggestions. I would like to ask the editorial office to combine the mentioned pictures 1,2,7 and 8 in one single figure.Formularende

Yes, I fully agree with the suggestion to enlarge our figures 3 and 4. But, at least, I am not able to realise this enlargement, this should be done by the editorial office, too. Thank you so much for this support!

Reviewer 3 Report

See attached

This is a good summary of the current state and will be useful to readers. I have a number of specific comments (see attached)

Author Response

Dear Reviewer,

Thank you very much for your reviews and the important hints and correction for improving the quality of our manuscript on Trichophyton intotineae.

Here our answers (in red):

Reviewer 3

This is a good summary of the current state and will be useful to readers. I have a number of specific comments (see attached)

More comments:

Both old and newer studies from India still report a mix of Trichophyton species associated with this outbreak. This includes a higher proportion T. rubrum cases than in your series. Do you think these other reports include misidentifications and, if not, why was T.rubrum superseded as the dominant species?

Answer:

In our own multicenter experience on tinea cruris, tinea corporis and tinea faciei in India, T. mentagrophytes was detectable in 138 (92.62 %) of all culture-positive skin samples. T. rubrum, however, was isolated in 11 (7.38 %) samples, only [20]. Similar results were obtained with a PCR-ELISA, 162 of 201 samples (80.56 %) were dermatophyte-positive. Of these, 151 (93.21 %) were identified as T. mentagrophytes and 11 (6.79 %) T. rubrum. Both old and newer studies from India still report a mix of Trichophyton species associated with this outbreak. This includes a higher proportion of T. rubrum cases than in our series. It is possible that these other reports include probably misidentifications due to morphological identification of dermatophytes. Our studies were based on molecular identification using sequencing of the DNA of all dermatophyte strains isolated.

The comment on endogenous cases is useful as it throws light on transmissibility. However, in which countries outside India and, from this paper, Germany has person to person transmission been documented. Cases that I have seen (5-6 per year) have all had an Indian connection

Is it possible to provide any estimate of the approximate timing of the mutation(s) which occurred with the emergence of T.indotineae?

Answer:

According to current knowledge, T. indotineae is mainly transmitted from person to person. Spread of T. indotineae infection as family case was documented from Iran [6] . None of the affected members in the family had history of travel to India and currently infection by the species is detected in many different provinces in Iran. But, also in Germany, intra-familiar transmission was proofen in at least one couple living in Germany, however, originating from Iraq [7]. A transmission of T. indotineae in Germany was documented in a by tinea corporis affected baby from Bahrain and his multiple family members, but also in a German woman and his husband from Saudi-Arabia, and in addition, from a Libyan to his female partner and their child.

The information regarding steroid combinations and this infection is important. Can you provide any information from outside India about regulating the use of potent steroid antifungal combinations. In some countries it is not possible by law to obtain these without prescription but are these countries the exception.

Answer:

Steroid combination creams like in India are available in a multitude of countries worldwide outside India, e.g. at the African continent, but also in Arab countries. In some countries, e.g. in Germany and other European countries, it is not possible by law to obtain these without prescription but are these countries the exception.

Page 14. Do you mean a reduced frequency of delayed hypersensitivity reactions ? As written it implies that this infection is associated with DTH.

These weak/reduced immunological reactions have been recorded and published previously with other forms of human dermatophytosis in contrast to models of infection. So, at present, are there any unique features of the immune response to indotineae? Or are we seeing a response that is similar to that occurring with other dermatophytes such as T.rubrum?

Answer:

Immunological characteristics of patients affected by chronic dermatophytosis include reduced interferon (IFN) γ, reduced Th1, IL-17-positive and Th17 cells as well as a impaired immune reaction (delayed type of hypersensitivity, DTH) in the intradermal test [60].

These weak/reduced immunological reactions have been recorded and published previously with other forms of human dermatophytosis in contrast to models of infection. At present, there are no unique features of the immune response to T. indotineae. The immunological response is similar to that occurring with other dermatophytes such as T. rubrum [61].